# Technology Opportunity Analysis Based on Machine Learning

**Junseok Lee** [1] , **Sangsung Park** [2,*] and **Juhyun Lee** [3,*]

1 Institute of Machine Learning and Big Data, Korea University, Seoul 02841, Republic of Korea
2 Department of Big Data Statistics, Cheongju University, Cheongju 28503, Republic of Korea
3 Institute of Engineering Research, Korea University, Seoul 02841, Republic of Korea
* Correspondence: hanyul@cju.ac.kr (S.P.); leeju@korea.ac.kr (J.L.)

**Abstract:** The sustainable growth of a company requires a differentiated research and development strategy through the discovery of technology opportunities. However, previous studies fell short of the need for utilizing outlier keywords, based on approaches from various perspectives, to discover technology opportunities. In this study, a technology opportunity discovery method utilizing outlier keywords is proposed. First, the collected patent data are divided into several subsets, and outlier keywords are derived using the W2V and LOF. The derived keywords are clustered through the K-means algorithm. Finally, the similarity between the clusters is evaluated to determine the cluster with the most similarity as a potential technology. In this study, 5679 cases of unmanned aerial vehicle (UAV) patent data were utilized, from which three technology opportunities were derived: UAV defense technology, UAV charging station technology, and UAV measurement precision improvement technology. The proposed method will contribute to discovering differentiated technology fields in advance using technologies with semantic differences and outlier keywords, in which the meaning of words is considered through W2V application.

**Keywords:** technology opportunity discovery; patent; word2vec; local outlier factor; UAV

## 1. Introduction

The development of differentiated core technology through continuous research and development (R&D) is essential for increased market competitiveness and the growth of a company. In R&D, the potential for technological progress is defined as a technology opportunity, the swift discovery and preemption of which are crucial [1,2]. The technology opportunity analysis (TOA) can facilitate strategic decisions on technology for decision makers and managers. Porter and Detampel (1995) previously proposed a TOA approach through a monitoring framework based on bibliometric analysis to discover differentiated technology opportunities [1]. Meanwhile, many studies have recently been conducted using intellectual property data to discover technology opportunities [3–10]. For example, Yoon and Park (2005) proposed a method to apply a morphology analysis to patents to identify technology opportunities [3]. In the proposed method, an analysis is performed by deriving keywords through text mining and by constructing a form matrix based on the keywords representing the characteristics of technology. Ma et al. (2013) proposed a TOA framework with an R&D analysis–competitor analysis–market analysis linking [4]. The proposed method extracts technical components through text mining and expert opinions and derives technology opportunities by comprehensively considering technology trends, secured national patent ratio by technology, and applicant ratio by technology. Lee et al. (2015) presented a patent map for the technology opportunity analysis [5]. They created a morphological patent context through text mining and morphological analysis and described a method for discovering novel patents by utilizing a local outlier factor (LOF) algorithm. Furthermore, they introduced a patent identification map, which

visualized patents using citations and the number of claims. Further, Song et al. (2017) proposed another method for discovering technology opportunities [6]. The concept of the proposed method is to find new technology opportunities through benchmarking by finding and presenting technologies with properties similar to the target technology through text mining for F-term, a Japanese patent classification system. Yoon and Magee (2018) proposed a patent map based on generative topographic mapping for technology opportunity discovery, and finally a TOA methodology through link prediction [7]. In the proposed method, keywords were extracted from patents to create GTM. Previous studies were typically based on the text-mining technique, deriving significant keywords for analysis. Ultimately, there is a risk of keywords with important meaning being removed from the analysis target owing to their low frequency, which may result in difficulties in discovering new opportunities. In other words, in statistical analysis, an outlier affects the result and needs to be removed. However, from a data mining point of view, outliers can be viewed as data that can provide new information. Therefore, the aim of this study is to introduce a method of utilizing outlier keywords from the viewpoint of discovering technology opportunities.

The proposed method is as follows. The collected patent data are divided into $n$ subsets, and text is extracted from each set. The data are refined through text preprocessing, such as removing unnecessary stopwords and punctuation marks from the extracted text data and integrating uppercase and lowercase letters. The refined data are converted into word embeddings through Word2Vec (W2V), and t-SNE is used to represent word vectors in a low-dimensional space. The LOF algorithm is applied to distinguish outlier keywords from keywords expressed in a low-dimensional space. The LOF calculated for each keyword that deviates from the interquartile range (IQR) is selected as an outlier keyword, and the selected keywords are clustered by utilizing the K-means algorithm to generate a technology cluster. Finally, a cluster with many similar clusters is defined as a potential technology with potential for development through similarity analysis between clusters.

The major characteristics of the method for technology opportunity discovery using outlier keywords proposed in this study are as follows. First, the collected data were divided into subsets, and each subset was analyzed according to the proposed method, as well as the similarity between the derived clusters in the last step. Thus, technologies with many similar clusters were selected as potential technologies to reduce the possibility of selecting the wrong technology. Second, the analysis was attempted considering the similarity of technology words. This approach may provide more useful information for analysts in deriving technical differentiation by considering the context.

Acronyms used in this paper are summarized in Table A1 (Appendix A). The remainder of this paper is organized as follows. Section 2 introduces the patent analysis with respect to technology management, as well as the machine-learning algorithm regarding W2V and the LOF. Section 3 introduces our study process in detail. Next, the results of the research are given in Section 4. Finally, the conclusions of this work and future work are presented in Section 5.

## 2. Background

### 2.1. Importance of Patent in Technology Management

In technology management, identifying technology trends and preparing in advance is crucial. For this purpose, research on intellectual property analytics (IPA) that analyzes patent big data through artificial intelligence has recently been conducted [11]. IPA is a technique that discovers information such as patterns or trends from intellectual property data for decision making. Ernst (2003) stated that technology management comprises management of technology creation, technology storage, and use of technical knowledge and that the functions of protection and information provision in patents correspond to this construction [12]. The applicant applies for a patent, and it is registered; they acquire the exclusive rights for the invention. Though they can protect their own invention, the

drawback is that the content of the invention is disclosed. Nevertheless, the advantage is that the invention is protected and can be used rightfully by the owner only. Therefore, many researchers and inventors apply for a patent. These unique characteristics of patents are sufficient for supporting Ernst's claim. Patents that provide information and protect inventions can also be considered technical documents. A patent document includes both structured and unstructured parts, which can be divided into (a) bibliographic information on technology, and (b) unstructured data related to technology [8]. This specificity of patents is applied to the following research fields in technology management: (a) discovery of promising technologies, (b) search for vacant technologies, (c) technology prediction, (d) analysis of technology trends, and (e) technology opportunity analysis [13,14]. Grzegorczyk and Glowinski (2020) introduced the concept of patent management strategy through literature review to secure sustainable technological competitiveness of companies in a fiercely competitive environment [15]. They conducted a literature review to divide patent management strategies into three categories: offensive patent strategy, defensive patent strategy, and leveraging patent strategy. Moreover, they emphasized that patent management is essential as competition becomes more intense in the high-tech field, while there is a general lack of related research.

### 2.2. Word2Vec

Words need to be vectorized for information processing using natural language, such as in text classification, sentiment analysis, and machine translation. Count-based representation is a method of vectorizing words by utilizing word frequency, term frequency–inverse document frequency (TF–IDF) score, and bag of words. Despite its wide applications, this method is disadvantageous in that it cannot reflect the position or meaning of words [16]. Word embedding is a method that can overcome the problems present in the count-based representation method, and the representative model is W2V, proposed by Mikolov et al. (2013) [17]. Two types of architectures were proposed for W2V as shown in Figure 1: the continuous bag-of-words (CBOW) model and continuous skip-gram model.

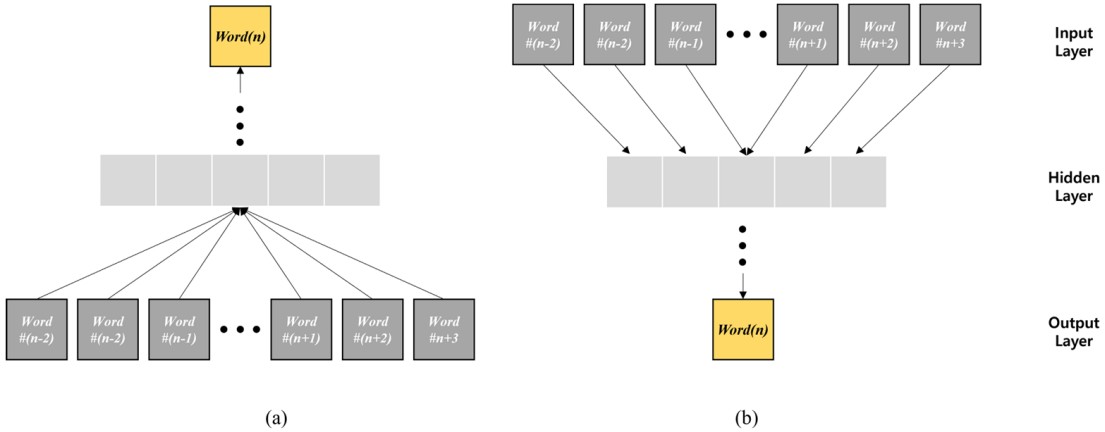

**Figure 1.** W2V architecture: (**a**) CBOW, (**b**) skip-gram [17].

The CBOW model uses a continuous distributed representation of the context. This model is characterized by predicting the current word (center word) in context. In particular, given the context, the missing word is predicted according to the window size. By comparison, the continuous skip-gram model, which is similar to CBOW, strives to maximize the classification of words based on other words in the same sentence. Because a word having low relevance to the current word is located farther than a related word, less weights are assigned to the farther-located words than nearby words by a smaller number of samples for the corresponding word.

*2.3. Local Outlier Factor*

The LOF algorithm, proposed by Breunig et al. (2000), is a density-based anomaly detection technique [18]. Outliers refer to data that deviate from the normal pattern among the observed data [19,20].

The mechanism for calculating LOF using the LOF algorithm is as follows. First, the distance to the *k*-th nearest neighbor excluding itself (*p*) is calculated, where the parameter *k* is a positive integer. As the *k* increases, more objects with similar reachability distances are included in the same neighborhood. The reachability distance is selected as a maximum among its *k*-distance of *o* or Euclidian distance between objects *p* and *o*, as represented in Equation (1).

$$Reachability\ distance_k(p,o) = max\{k\ distance\ of\ o,\ d(p,o)\} \tag{1}$$

Thus, if there are many other objects near object *p*, *k*-distance of *o* and *d*(*p*, *o*) are similar, but if there are few or no other objects around object *p*, *d*(*p*, *o*) then *k*-distance of *o* will be large.

Local reachability distance (*lrd*) means density around the object *p*. Therefore, *lrd* is represented as shown in Equation (2), where $N_k(p)$ is a set of objects within *k*-th nearest object for object *p*.

$$lrd_k(p) = \left( \frac{\sum Reachability\ distance_k(p,o)}{|N_k(p)|} \right)^{-1} \tag{2}$$

Finally, the LOF with respect to the object *p*, can be calculated through Equation (3), where LOF refers to the average of rations of local reachability distance with respect to the object *p* and *o*.

$$\text{LOF}_k(p) = \frac{\sum \frac{lrd_k(o)}{lrd_k(p)}}{|N_k(p)|} \tag{3}$$

As shown in Equation (3), LOF(*p*) is the average of the ratio of the local reachability density of *p* to the local reachability density of neighbor *o*. Thus, inliers have approximately 1, and outliers have values far from 1. Figure 2 shows the cases for object *p* as an inlier and outlier.

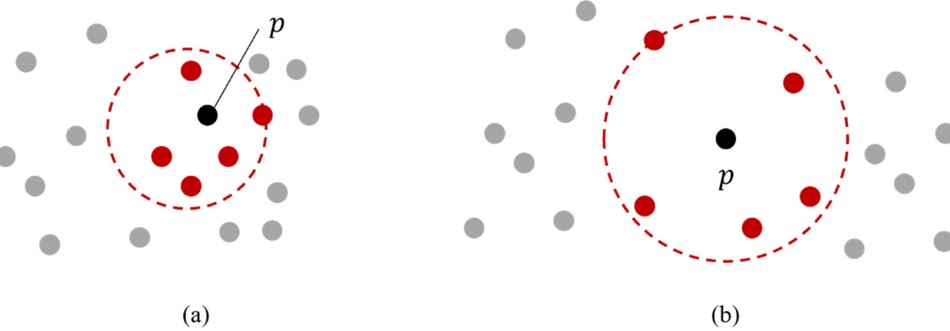

(a)  (b)

**Figure 2.** Example of (**a**) *p* is an inlier (**b**) *p* is an outlier where k is set to 5.

LOF is often utilized for identifying novelty [5,9,21]. Lee et al. (2015) used LOF for identifying the novelty of patents [5]. Moreover, Jeon et al. (2022) aimed to apply LOF to a document vectorized through doc2vec to determine the novelty of a patent and further proposed a novelty score [21]. Further, Choi et al. (2022) proposed an approach using a language model and LOF for business opportunity analysis [9]. The present study utilizes the abovementioned values for discovering outlier keywords with novelty to discover technology opportunities.

## 3. Methodology

### 3.1. Introduce to Methodology

The proposed methodology—the technology opportunity analysis based on machine learning—is described in this section. Figure 3 shows the concept for the designed methodology and Figure 4 shows our research process.

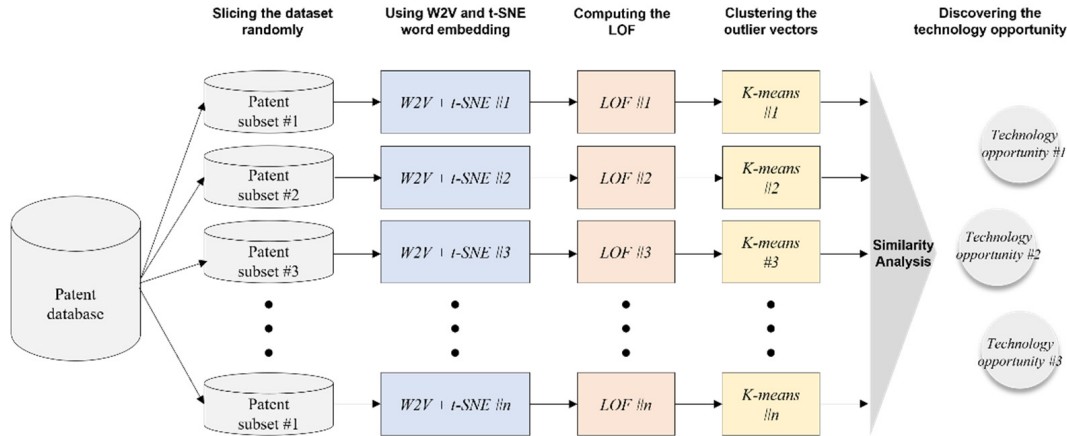

**Figure 3.** Concept diagram for the proposed methodology.

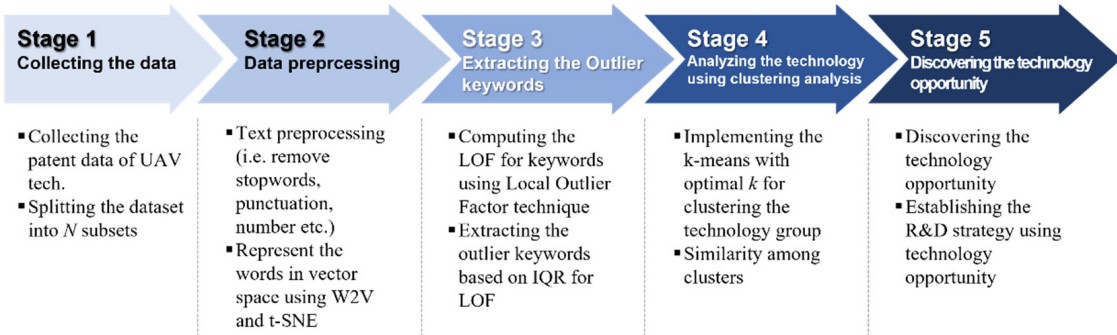

**Figure 4.** Proposed methodology process.

(Step 1) Collecting the patent data of specific technology:

- Collecting the data of the drone technology field from the patent database;
- Randomly splitting the patent dataset into subsets.

(Step 2) Splitting the patent dataset into subsets and representing the words in vector space:

- Creating the corpus;
- Cleaning the text data using text preprocessing technique, for example, removing punctuation, stopwords, and numbers;
- Constructing the document-term matrix (DTM);
- Applying DTM to W2V words represented in vector space;
- Reducing the dimensionality from high-dimensional to two-dimensional using t-SNE algorithm.

(Step 3) Modeling for detecting the outlier keyword using LOF:

- Estimating the anomaly score using the LOF technique;
- Extracting the outlier keywords using IQR for each subset.

(Step 4) Clustering outlier keywords using the K-means algorithm:

- Checking the elbow point from 1 to 10 and selecting the optimal *k* for each subset;
- Clustering outlier keywords for each subset.
- (Step 5) Similarity analysis and discovering the technology opportunity:

- Analyzing the similarity between clusters;
- Identifying the technology opportunity through results of similarity analysis and establishing the R&D strategy.

### 3.2. Collecting the Data—Patent Data Related to UAV

The purpose of the present study is to propose a method to find important technical information from outliers based on the scientific method so that it can be used in R&D strategy establishment. In this study, we intended to discover technology opportunities by applying the proposed method to the unmanned aerial vehicle (UAV) technology field, which is a representative high-tech field. Patent data were employed to apply the proposed method. A patent is a technical document. Therefore, in this work, we used patent data to explain the proposed method. Table 1 presents the information for collected patents.

**Table 1.** Collected patent data.

| Technology Field | Number of Documents | Patent DB |
|:---:|:---:|:---:|
| UAV | 5679 | WISDOMAIN |

For our study, we collected patents on UAV-related technology from WISDOMAIN, which is a patent database provider. The collected patent data included 5679 patents applied in the US during 2000–2021. A patent includes various types of information, such as the title, abstract, and claims. In this study, the main subject of analysis was the text. Therefore, text information from patents, titles, abstracts, and claims were used.

### 3.3. Data Preprcessing

In this stage, the text data were cleaned to conclude the precise analysis result. The patent document is written grammar, spelling, and well expression. However, as mentioned above, several sentences include unimportant information, such as punctuation, numbers, and stopwords. A stopword denotes a word that is not necessary for analysis, for example, "I", "by", and "to." They do not affect the result, cannot positively influence the result of the analysis, and can increase the computation complexity. Therefore, they need to be removed before analysis. In addition, the point of text cleaning is word form. Although words may be synonymous but not the same part of speech, the feature is recognized as another form in text mining. For example, "means" and "mean" have the same meaning; however, because the subject is not the same, they are written in different forms. These add to the problem of sparsity and make analysis difficult. Therefore, to avoid this problem, stemming needs to be used in general. It is effective in text mining, but identifying the meaning only based on stemming is sometimes difficult. Accordingly, a lemma was extracted using WordNet Lemmatizer from the NLTK package of Python in our experiment, and only their nouns were extracted.

After text preprocessing, the words were embedded as vectors using Word2Vec. The W2V hyperparameters are given in Table 2.

**Table 2.** W2V hyperparameters.

| Hyperparameter | Candidates |
|:---:|:---:|
| Vector Size | 100 |
| Window | 3 |
| Algorithm | Skip-gram |

The preprocessed text data had high dimensionality. Accordingly, using W2V, they were transformed into 100-dimensional data. Although the dimension was reduced, the representation visuality was insufficient for applying LOF. Therefore, the embedded vectors

were represented in 2D space by applying the t-SNE algorithm [22]. Table 3 presents the t-SNE hyperparameters used. The described models were applied to each subset.

**Table 3.** t-SNE hyperparameters.

| Hyperparameter | Candidates |
| --- | --- |
| N_component | 2 |
| Initialization of embedding | PCA |

### 3.4. Extracting the Outlier Keywords Using Local Outlier Factor

The aim of this study is to identify the technology opportunity on target technology from outlier keywords using machine learning techniques. The data were processed into an analytic form via the text-preprocessing process described in Section 2. In this study, we attempted to apply the LOF algorithm to determine the outliers. The hyperparameters used to compute the LOF are listed in Table 4.

**Table 4.** LOF algorithm hyperparameters.

| Hyperparameter | Candidates |
| --- | --- |
| n_neighbors | 20 |
| metric | minkowski |

The LOF can be calculated using the LOF algorithm described in Section 2. In this study, the final outlier keyword was selected by applying IQR Rules to the LOF. The IQR is a range that is a difference between the third quantile $Q_3$ and the first quantile $Q_1$ of a dataset. The formula for calculating the IQR is as follows Equation (4):

$$IQR = Q_3 - Q_1 \tag{4}$$

Thus, if the LOF of the keyword is smaller than $Q_1 - 1.5 \times IQR$ or larger than $Q_3 + 1.5 \times IQR$, it is selected as an outlier keyword. The final outlier keyword is extracted using the IQR calculated for each subset.

### 3.5. Analyzing the Technology Using Clustering Technique

This stage extracts and analyzes the technology information from outlier keywords. To derive the technology opportunity, the K-means clustering algorithm [23–25], which is one of the unsupervised machine learning techniques, is applied in the proposed methodology. This algorithm aims to divide a given dataset into a set of *k* clusters. The algorithm process is shown in Figure 5.

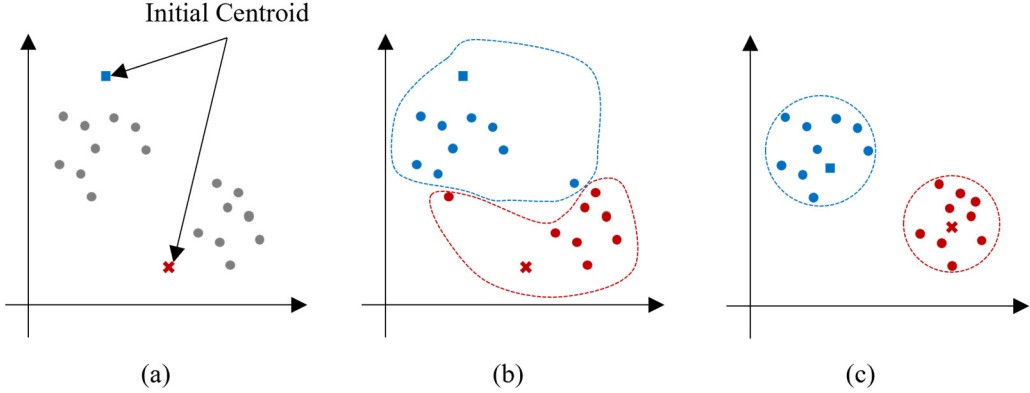

**Figure 5.** K-means clustering algorithm process: (**a**) initial step, (**b**) centroid update step, (**c**) completed step.

In the first step, the *k* objects, which is the mean or centroids of a cluster, are placed in vector space randomly. As shown in (2) of Figure 5, each of the remaining objects are assigned to a closet centroid, where the closet is calculated as the Euclidean distance between the centroid and an object as in Equation (5).

$$\text{dist}(x, c) = \sqrt{\sum_{i=1}^{n}(x_i - c)^2} \tag{5}$$

where *c* means the centroid of each cluster. Next, the location of centroids is updated and minimizes the within-cluster sum of squares (WCSS) between the object and centroid, continuously. This step is repeated until the centroid is not changed. If the centroid changes no more, the algorithm is stopped. The objective of this algorithm is represented as Equation (6).

$$\underset{S}{\text{argmin}} \sum_{i=1}^{k} \sum_{x \in S_i} dist(x, c_i)^2 \tag{6}$$

To find the optimal *K*, the elbow method based on the WCSS is used in our experiment. It is one of the methods for finding the optimal *k*. According to the *k*, compute the WCSS, and then choose the *k* for which it first starts to diminish. This point is considered optimal *k*.

This algorithm plays a role that helps to identify the technology information from the outlier keyword sets.

### 3.6. Discovering the Technology Opportunity

The potential technology opportunities are extracted by the k-means algorithm. In order to identify the most important technology opportunities, the proposed methodology uses similarity analysis. The similarity analysis is the simplest and more powerful method to find similarities [23,24]. Equation (7) shows the cosine similarity analysis.

$$similarity(x, y) = \frac{x \cdot y}{\|x\| \|y\|} \tag{7}$$

Finally, a similarity analysis between each pair of clusters was conducted to determine a technology cluster with high similarity values as a field with a technology opportunity. A cluster that has a higher similarity average than others is considered a technology opportunity.

### 4. Experiments and Results

This section describes the experimental process and results. Figure 6 presents the data that were previously collected.

Figure 6 shows the collected data on the number of patent applications, showing a low number of applications from 2000 to 2015. However, a sharp increase in the number of applications is seen from 2016, which reaches the highest in 2018.

In this study, word embedding was performed using W2V, as described in Section 3.2 and 3.3, to represent the words of the collected UAV patent data in word vector space, rather than those that were determined to be outliers because of their relatively low frequency compared to other keywords. A text refinement process proceeds for embedding. Sentences comprise unnecessary symbols, numbers, and punctuation marks in addition to letters. For constructing sentences, there are various forms of words, such as nouns, verbs, and adjectives according to grammar. However, punctuation marks, symbols, general verbs, and modifiers have no significant effect on the analysis. Thus, we utilized WordNet Lemmatizer and part-of-speech tagging (POS tagging) in NLTK to best restore words to their original form and extracted nouns alone. POS tagging, as shown in Figure 7, identifies the parts-of-speech of words in sentences and adds tags.

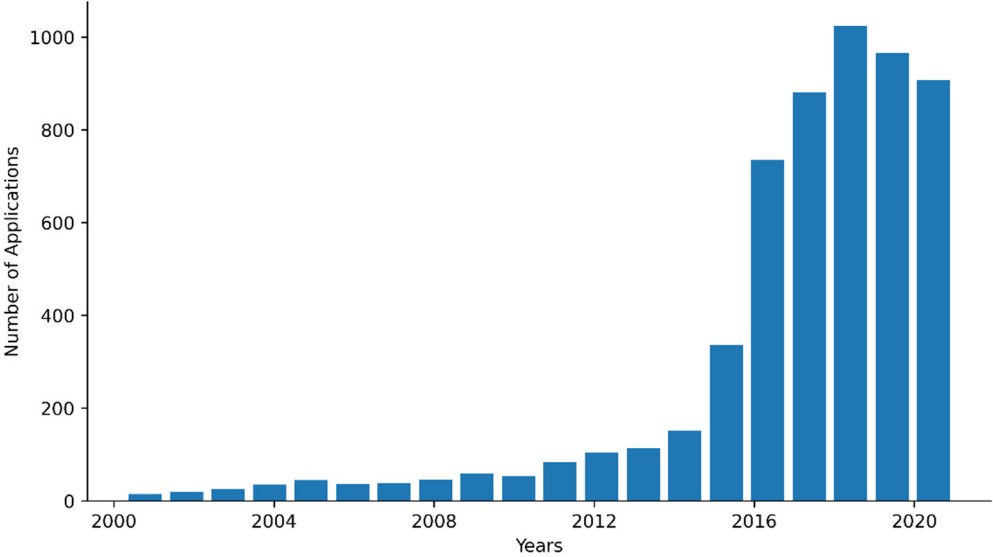

**Figure 6.** Collected UAV patent data.

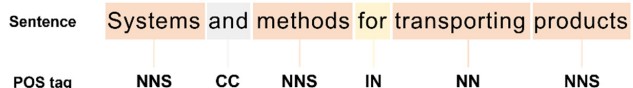

**Figure 7.** Example of POS tagging.

The t-SNE algorithm was applied to the refined text data to represent the 100-dimensional embedded words in 2D space. Figure 8 shows some keywords of the subset in two dimensions.

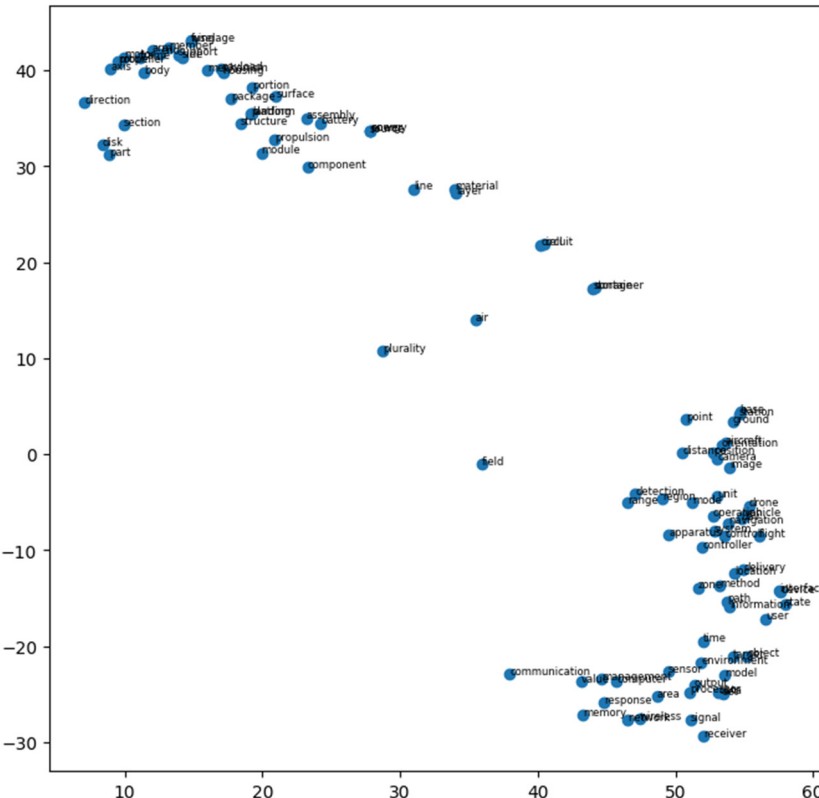

**Figure 8.** Example of word embedding results.

The LOF algorithm was applied to calculate the LOF for each word. Table 5 presents the corresponding results. The LOFs of 20,652 words could be obtained from a total of 10 subsets. Table 6 and Figure 9 show the basic statistics for the LOF of each subset.

**Table 5.** Sample for LOF of keywords.

| Word | Subset No. | X | Y | LOF |
|---|---|---|---|---|
| vehicle | 0 | 55.023 | −6.405 | 1.088 |
| drone | 0 | 55.342 | −5.365 | 1.093 |
| system | 0 | 52.801 | −8.032 | 1.033 |
| device | 1 | 57.626 | −0.078 | 0.963 |
| localizing | 1 | −59.189 | −27.479 | 1.103 |
| . . . | . . . | . . . | . . . | . . . |
| traffic | 7 | 47.543 | 17.132 | 1.344 |
| part | 7 | 25.866 | 36.117 | 1.482 |
| station | 8 | 54.892 | −17.796 | 1.465 |
| receiving | 8 | 28.824 | 26.138 | 0.938 |
| tag | 9 | 35.094 | 7.839 | 1.448 |

**Table 6.** Descriptive statistics for keywords of subsets.

| Subset No. | Mean | Std. | Min | Max | Q1 | Q3 | IQR |
|---|---|---|---|---|---|---|---|
| 0 | 1.02698 | 0.061575 | 0.936517 | 1.453869 | 0.987785 | 1.043947 | 0.056163 |
| 1 | 1.021218 | 0.053211 | 0.942201 | 1.448178 | 0.989804 | 1.034662 | 0.044858 |
| 2 | 1.028386 | 0.06783 | 0.936725 | 1.608826 | 0.989202 | 1.042044 | 0.052841 |
| 3 | 1.0243 | 0.055879 | 0.934708 | 1.45417 | 0.988664 | 1.043112 | 0.054447 |
| 4 | 1.026969 | 0.058455 | 0.937783 | 1.439178 | 0.990205 | 1.047507 | 0.057302 |
| 5 | 1.024395 | 0.063914 | 0.936138 | 1.640614 | 0.989406 | 1.035572 | 0.046166 |
| 6 | 1.026492 | 0.061354 | 0.931228 | 1.515098 | 0.987773 | 1.042835 | 0.055062 |
| 7 | 1.02419 | 0.058558 | 0.941442 | 1.570052 | 0.988169 | 1.039446 | 0.051278 |
| 8 | 1.028224 | 0.064822 | 0.927807 | 1.536986 | 0.988537 | 1.044326 | 0.055789 |
| 9 | 1.029998 | 0.068492 | 0.931828 | 1.448094 | 0.98459 | 1.052209 | 0.067619 |

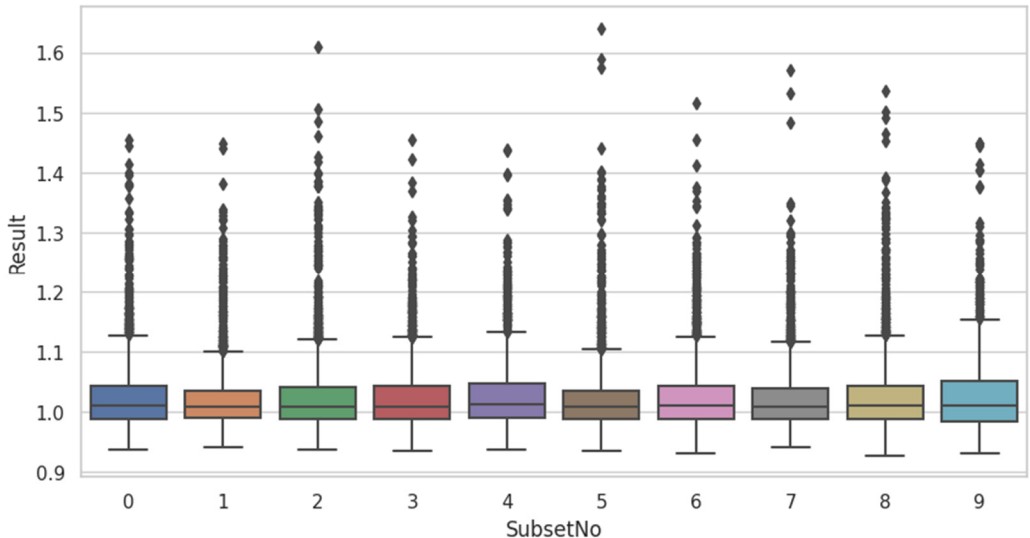

**Figure 9.** Boxplot of LOF of subsets.

The IQR was utilized to select the final outlier keyword. Accordingly, a total of 1390 keywords, including duplicates, were determined to be outliers. The keywords extracted from each subset were clustered using the K-means algorithm to identify the

technology information indicated by these outliers. In this study, the elbow method was utilized to determine the optimal number of clusters, *k*. To apply the elbow method, the WCSS is used as a measurement. Figure 10 shows plots to identify the elbow points, and Table 7 presents the optimal number of clusters, *k*, for each subset.

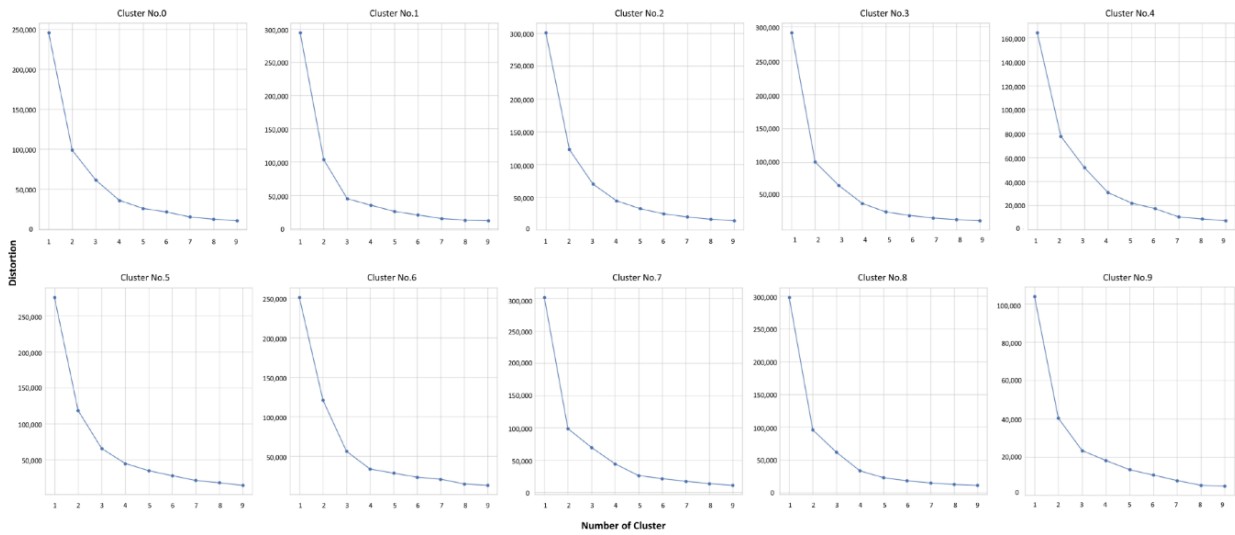

**Figure 10.** Plot for within-cluster-sum of squared errors for *k*.

**Table 7.** Optimal *k* for each subset.

| Subset 0. | Subset 1. | Subset 2. | Subset 3. | Subset 4. | Subset 5. | Subset 6. | Subset 7. | Subset 8. | Subset 9. |
|-----------|-----------|-----------|-----------|-----------|-----------|-----------|-----------|-----------|-----------|
| 4 | 3 | 4 | 4 | 4 | 4 | 4 | 5 | 4 | 2 |

The results of the elbow points revealed that a total of 38 clusters could be created. Further, a similarity analysis among clusters was conducted to find the hidden technology field. The similarity analysis utilized cosine similarity. The results of the similarity analysis among 38 clusters were presented using a heat map, as shown in Figure 11.

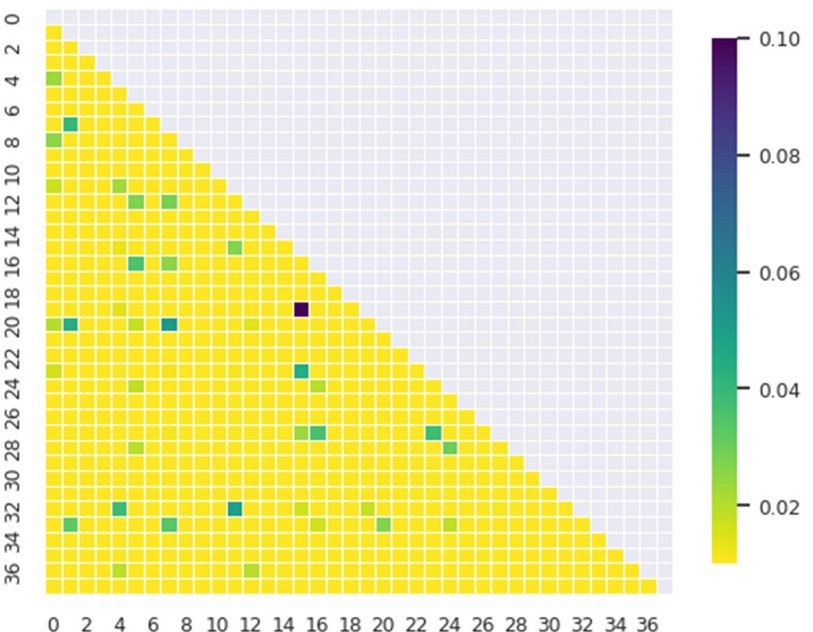

**Figure 11.** Heat map for similarity analysis.

The similarity analysis utilized to determine the similarity among the 38 clusters revealed that the average similarity among clusters 7, 20, and 16 was high. Keywords were identified, and technology definitions were realized for the top three clusters with high average similarity. Table 8 presents the average similarity for each cluster, and Table 9 presents keywords and technology definitions for major clusters.

**Table 8.** Similarity average for each cluster.

| Subset Index | Cluster Index | Representative Index | Similarity Average |
|---|---|---|---|
| 0 | 0 | 0 | 0.057 |
| | 1 | 1 | 0.059 |
| | 2 | 2 | 0.034 |
| | 3 | 3 | 0.033 |
| 1 | 0 | 4 | 0.065 |
| | 1 | 5 | 0.058 |
| | 2 | 6 | 0.031 |
| 2 | 0 | 7 | 0.079 |
| | 1 | 8 | 0.045 |
| | 2 | 9 | 0.034 |
| | 3 | 10 | 0.029 |
| 3 | 0 | 11 | 0.059 |
| | 1 | 12 | 0.054 |
| | 2 | 13 | 0.034 |
| | 3 | 14 | 0.037 |
| 4 | 0 | 15 | 0.057 |
| | 1 | 16 | 0.066 |
| | 2 | 17 | 0.032 |
| | 3 | 18 | 0.031 |
| 5 | 0 | 19 | 0.054 |
| | 1 | 20 | 0.069 |
| | 2 | 21 | 0.035 |
| | 3 | 22 | 0.029 |
| 6 | 0 | 23 | 0.061 |
| | 1 | 24 | 0.058 |
| | 2 | 25 | 0.030 |
| | 3 | 26 | 0.035 |
| 7 | 0 | 27 | 0.066 |
| | 1 | 28 | 0.051 |
| | 2 | 29 | 0.041 |
| | 3 | 30 | 0.029 |
| | 4 | 31 | 0.031 |
| 8 | 0 | 32 | 0.061 |
| | 1 | 33 | 0.057 |
| | 2 | 34 | 0.031 |
| | 3 | 35 | 0.031 |
| 9 | 0 | 36 | 0.049 |
| | 1 | 37 | 0.038 |

**Table 9.** Keywords and technology definition for major clusters.

| Cluster No. | Keyword | Technology Definition |
|:---:|:---:|:---:|
| 7 (2-0) | phenomenon, drones, transformer, intent, oscillator, inhouse, lag, holding, rectifier, interdiction, concern, multi trigger, low rate, analyzing, sram, throttle, dc, diversity, nonoverlapping, disengagement, prevent, po, practice, dsl, constellation, system one, extremity, exporting, firefighting, multitab, display, derivatives | UAV defense technology |
| 20 (5-1) | power, air, surface, base, station, platform, frame, housing, ground, direction, axis, battery, center, source, material, channel, rotation, space, angle, layer, element, weight, plane, contact, block, prepreg, sensory, gate, ply, plate, synchronization, metal, cradle, cue, hanger, av, substrate, oblique, mobility, glide, vias, system | UAV charging station(platform) technology |
| 16 (4-1) | stack, ml, nonce, root, mask, magnetization, exhibition, interrogation, police, charging, dummy, provider, foam, redistribution, passivation, threat, heathing, reinforcement, operate, inhibitor, bounding, timestamp, dial, rearend, tracer, releasing, signaltonoise, cuav, spotlight, cone, metaloxide, molding, send, ingestion, pipeline, merkel, surrounding, higherlevel, fit, cap, metallization, blend | To improve the UAV measurement technology |

Clustering outlier keywords using K-means clustering indicated that there were many clusters similar to the clusters in Table 9. The keywords included in the three clusters revealed the following potential technologies: (1) UAV defense technology, (2) UAV charging station technology, and (3) UAV measurement precision improvement technology. Thus, based on the above results, important findings can be identified. First, technologies related to defense, charging, and measurement precision improvement can be determined as technology opportunities in the UAV field, owing to a relative lack of technological development. Second, our method can be applied to actual patent data. In other words, this experiment shows that extracting information is possible from outliers. Therefore, we are able to recognize the important fact that outliers can be usefully utilized.

## 5. Conclusions and Future Works

In this study, we proposed a method for discovering technology opportunities utilizing outlier keywords. Outliers typically occur because of mismeasured values or abnormal behavior. The anomaly detection aims to discover patterns found in outliers and utilize them as significant information. Previous studies explored technology predictions, promising technology extractions, and a vacant technology search methodology using major keywords. However, the keyword outliers could be attributed with relatively less frequency than that of other keywords, resulting in statistically classified outliers, rather than erroneous entries or abnormal behavior. Thus, there is a need for the cautious removal of outliers in data analysis using keywords because of the high likelihood of a loss of significant information if the outlier is unconditionally excluded from the analysis.

Through this study, we intend to present a method for discovering technology opportunities by utilizing outlier keywords. To validate the proposed methodology, a case study was conducted using UAV patent data, a new technology field. In this study, texts are targeted. Thus, the dataset comprises titles, summaries, and claims from patent documents. The constructed dataset is structured through data cleansing and preprocessing. The structured text data are difficult to analyze as high-dimensional data. Thus, word embedding and dimension reduction are performed using W2V and t-SNE. The LOF algorithm is further applied to find outlier keywords among data expressed in a 2D vector space. The final outlier keywords are selected by applying the IQR method to the anomaly score obtained through the LOF. Finally, clustering is performed by applying the K-means

algorithm to derive significant meanings among the selected outlier keywords. The analysis of the UAV patent data collected through the proposed method revealed that the following technologies were potential technologies: (1) UAV defense technology, (2) UAV charging station technology, and (3) UAV measurement precision improvement technology. From the perspective of the technology opportunity, the abovementioned technologies can be interpreted as follows. There may be a relative lack of research on the technology information derived through analysis among other UAV technology studies derived from outliers. Therefore, these technology fields can be considered to have sufficient potential for development.

In this study, we present a methodology to discover technology opportunities based on scientific methods, with the following contributions. In the count-based word representation method, even though the meaning is similar, it is sometimes divided as an outlier due to a difference in frequency, resulting in a difficulty in interpreting the meaning. However, the proposed method utilizes W2V to select outlier keywords. This approach is meaningful in that the positional information of the words is used to semantically divide the main technology and the technology containing outlier keywords. Second, this study is meaningful in that it presents a method that can discover potential technologies with potential for development through outlier keywords from the perspective of technology opportunities. This method will significantly contribute to differentiated technology development strategies. Technological development is also related to changes in time and society. However, the proposed method has a limitation in that it omits and leaves time series factors in the analysis up to the analyst. In addition, this study has a limitation in that it was analyzed with interest only in the contents of the document. Therefore, to promote technology intelligence activities based on scientific methodologies, there is a need to present a method that considers various factors, such as financial, society, and market, in future research, which can consider technological development.

**Author Contributions:** J.L. (Junseok Lee) conceptualized and designed this research and conducted the experiment as described. S.P. collected the data set for the experiment. J.L. (Juhyun Lee) analyzed the data to show the validity of this paper. In addition, all authors have co-operated with each other in revising the paper. All authors have read and agreed to the published version of the manuscripts.

**Funding:** This research was supported by the MOTIE (Ministry of Trade, Industry, and Energy) in Korea, under the Human Resource Development Program for Industrial Innovation (Global) (P0017311), supervised by the Korea Institute for Advancement of Technology (KIAT). This work was supported by the National Research Foundation of Korea (NRF) grant, funded by the Korean government (MSIT) (No. NRF-2020R1A2C1005918). This research was supported by the Basic Science Research Program through the National Research Foundation of Korea (NRF), funded by the Ministry of Education (No. NRF-2022R1I1A1A01069422).

**Informed Consent Statement:** Not applicable.

**Data Availability Statement:** Not applicable.

**Conflicts of Interest:** The authors declare no conflict of interest.

## Appendix A

**Table A1.** List of Acronyms.

| Acronyms | Explanation |
| --- | --- |
| W2V | Word2Vec |
| LOF | Local outlier factor |
| UAV | Unmanned aerial vehicle |

**Table A1.** *Cont.*

| Acronyms | Explanation |
|---|---|
| R&D | Research and Development |
| TOA | Technology opportunity analysis |
| GTM | Generative topology mapping |
| IQR | Interquartile range |
| IPA | Intellectual property analytics |
| TF-IDF | Term frequency-Inverse document frequency |
| CBOW | Continuous bag-of-words |
| DTM | Document-Term matrix |
| t-SNE | t-distributed stochastic neighbor embedding |
| PCA | Principal component analysis |
| POS | Part-of-speech |

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
