# Peer review of "Technology Opportunity Analysis Based on Machine Learning"

_axioms, doi:10.3390/axioms11120708_

Round 1
Reviewer 1 Report
COMMENTS:
1. Need List of Acronyms
2. Very interesting paper
3. Very well organized and presented
4. Novel idea of the use of traditional processing techniques in Text Analysis, Statistics, Pattern Recognitio and Machine Learning
5. Time Series analysis would be valuable specifically for time tagging relevant patent information and forcasting/prediction as is done in estimation/control and financial analysis
See attached file for other comments/suggestions. PDF was converted to WORD and edited using Track Changes

Reviewer 2 Report
The article needs the following changes:
1) What problem was studied and why is it important?
2) What methods were used?
3) What are the important results?
4) What conclusions can be drawn from the results?
5) What is the novelty of the work and where does it go beyond previous efforts in the literature?
6) What is the main contribution of this paper?
7) What does the current paper add to the subject area compared with other published studies?
8) I think the results of the paper pave the way to new avenues that are fully awaited. Therefore, future works should be added in the conclusion part.
9) The changes made based on the comments should be written in color.
After carrying out these changes, I recommend that the paper can be published.
Round 2
Reviewer 2 Report
The paper can be accepted